# Effects of LPS from *Rhodobacter sphaeroides*, a Purple Non-Sulfur Bacterium (PNSB), on the Gene Expression of Rice Root

**DOI:** 10.3390/microorganisms11071676

**Published:** 2023-06-28

**Authors:** Ranko Iwai, Shunta Uchida, Sayaka Yamaguchi, Daiki Nagata, Aoi Koga, Shuhei Hayashi, Shinjiro Yamamoto, Hitoshi Miyasaka

**Affiliations:** 1Department of Applied Life Science, Sojo University, 4-22-1 Ikeda, Nishiku, Kumamoto 860-0082, Japanshayashi@bio.sojo-u.ac.jp (S.H.); syamamot@bio.sojo-u.ac.jp (S.Y.); 2Ciamo Co., Ltd., G-2F Sojo University, 4-22-1 Ikeda, Nishiku, Kumamoto 860-0082, Japan; aoi_koga@ciamo.co.jp

**Keywords:** purple non-sulfur bacterium, PNSB, LPS, *Rhodobacter sphaeroides*, RNA-seq

## Abstract

The effects of lipopolysaccharide (LPS) from *Rhodobacter sphaeroides*, a purple non-sulfur bacterium (PNSB), on the gene expression of the root of rice (*Oryza sativa*) were investigated by next generation sequencing (NGS) RNA-seq analysis. The rice seeds were germinated on agar plates containing 10 pg/mL of LPS from *Rhodobacter sphaeroides* NBRC 12203 (type culture). Three days after germination, RNA samples were extracted from the roots and analyzed by RNA-seq. The effects of dead (killed) PNSB cells of *R. sphaeroides* NBRC 12203^T^ at the concentration of 10^1^ cfu/mL (ca. 50 pg cell dry weight/mL) were also examined. Clean reads of NGS were mapped to rice genome (number of transcript ID: 44785), and differentially expressed genes were analyzed by DEGs. As a result of DEG analysis, 300 and 128 genes, and 86 and 8 genes were significantly up- and down-regulated by LPS and dead cells of PNSB, respectively. The plot of logFC (fold change) values of the up-regulated genes of LPS and PNSB dead cells showed a significant positive relationship (r^2^ = 0.6333, *p* < 0.0001), indicating that most of the effects of dead cell were attributed to those of LPS. Many genes related to tolerance against biotic (fungal and bacterial pathogens) and abiotic (cold, drought, and high salinity) stresses were up-regulated, and the most strikingly up-regulated genes were those involved in the jasmonate signaling pathway, and the genes of chalcone synthase isozymes, indicating that PNSB induced defense response against biotic and abiotic stresses via the jasmonate signaling pathway, despite the non-pathogenicity of PNSB.

## 1. Introduction

Purple non-sulfur bacteria (PNSB) are one of the various phototrophic microorganisms [1,2]. PNSB have photosynthetic reaction centers and light-absorbing pigments (bacteriochlorophyll and carotenoids) to convert light energy into chemical energy (ATP). Unlike cyanobacteria, PNSB do not use water as an electron donor for photochemical reaction, and do not emit oxygen. This type of photosynthesis is called anoxygenic photosynthesis. PNSB are facultative anaerobes and relatively easy to cultivate. PNSB are Gram-negative bacteria. Gram-negative bacteria are sometimes pathogenic, but to our knowledge, there has been no report of pathogenic PNSB and there have been no reports of opportunistic infection by PNSB in humans. PNSB have a wide variety of biotechnological applications in agriculture [3,4,5], aquaculture [6,7], biomaterial production [8,9,10], renewable energy production [11,12], wastewater treatment [13], and bioremediation [14]. 

The most widespread application of PNSB is the promotion of plant growth and the improvement of the quality of food crops. The mechanisms for the promotion of plant growth by the PNSB have been the subject of intensive studies [3,15], and the mechanisms can be classified under (1) the biological activities of the PNSB and (2) the production of various beneficial (promoting plant growth) substances by the PNSB. The biological activities include nitrogen fixation, solubilization of soil phosphate, and degradation of toxic hydrogen sulfide. PNSB also suppresses the generation of methane, a greenhouse gas, from paddy fields by competing organic substrates with methanogenic bacteria [16]. The beneficial substances produced by PNSB include polyphosphate, pigments, vitamins, and plant growth-promoting substances (PGPSs), such as indole-3-acetic acid (IAA), a plant hormone, and 5-aminolevulinic acid (ALA). ALA is a plant growth regulator that assists in plant growth and yield and alleviates various abiotic stresses [17].

In addition to these aforementioned mechanisms, we recently proposed lipopolysaccharide (LPS), a component of the Gram-negative bacterial cell wall, as a new active ingredient in PNSB [18,19]. Hayashi et al. [18] reported the growth-promoting effect of lipopolysaccharide (LPS) from PNSB in plants for the first time, and the effective concentration of LPS from *Rhodobacter sphaeroides* NBRC 12203^T^ was 10 pg/mL. Iwai et al. [19] reported that biopriming of rice seeds by LPS from *R. sphaeroides* NBRC 12203^T^ at a concentration of 5 ng/mL improved the root growth of rice seedlings, providing further evidence of the effectiveness of LPS from PNSB in plants. In mammals, LPS acts as an endotoxin, causing inflammatory responses through a TLR4-like signaling pathway at concentrations ranging from pg/mL to ng/mL. In plants, the receptor for LPS, known as lipooligosaccharide-specific reduced elicitation (LORE), has been identified [20], but compared to mammals, plants are believed to be much less sensitive to LPS [21]. There have been many reports on the effects of LPSs on plant cells and tissues, and these effects include reactive oxygen species (ROS) generation [22,23,24], induction of secondary metabolites production [25], induction of resistance to plant pathogens, such as fungi [26] and nematode [27], and growth promotion [28]. In all of these previous studies reporting the effects of LPSs from various Gram-negative bacteria on plants, the effective LPS concentration was 10 to 100 μg/mL or higher, which was a thousand to a million times higher than the effective concentration range (pg/mL to ng/mL) in mammals. Thus the much lower effective concentration of LPSs from *R. sphaeroides* NBRC 12203^T^ in plants than those of other various Gram-negative bacteria is of great interest in terms of biotechnological application of PNSB in agriculture. Additionally of great interest, in terms of basic biology, for example, what kind of genes are induced by LPSs from PNSB, and how different (or similar) are they to those induced by the other members of LPSs. In this study, to better understand the effects of LPS from *R. sphaeroides* NBRC 12203^T^ in plants, we examined its effects on the gene expression of the roots of rice by RNA-seq analyses.

## 2. Materials and Methods

### 2.1. Rice Cultivar

A rice (*Oryza sativa*) cultivar, Akita-komachi, was used, and the seeds were purchased from Noken Co., Ltd., Kyoto, Japan.

### 2.2. PNSB Strain and LPS

*Rhodobacter sphaeroides* NBRC 12203^T^, the type culture of *Rhodobacter sphaeroides*, was obtained from the NBRC (Biological Resource Center, NITE, Kisarazu, Japan) culture collection. LPS of *R. sphaeroides* ATCC 17,023 [29] was purchased from InvivoGen (San Diego, CA, USA). *R. sphaeroides* NBRC 12203^T^ and *R. sphaeroides* ATCC 17,023 are the identical strains with different culture collection numbers. To avoid confusion, in the present paper, we call this LPS as LPS from *R. sphaeroides* NBRC 12203^T^.

For the preparation of dead (killed) PNSB cells, the cells were collected by centrifugation, and the cell pellet was dried at 60 °C for 6 h. Distilled water was added to the dried cell pellet, and the pellet was disrupted by sonication.

### 2.3. Cultivation of Rice Seedlings on Agar Plates

Rice seeds were surface sterilized by immersion in 70% (*v*/*v*) ethanol for 2 min followed by 2% sodium hypochlorite for 20 min. The seeds were germinated on no nutrient agar (0.8%) square plates (140 mm × 100 mm × 14.5 mm, Tokyo Garasu Kikai Co., Ltd., Tokyo, Japan), and the seedlings were grown on vertically oriented plates at 28 °C in the dark.

### 2.4. Analysis of Root Development by WinRhizo Image Analyzing System

Analysis of root development by WinRhizo image analyzing system was carried out as described previously [19]. Root images were acquired using a scanner (Epson GT-X970, Seiko Epson Co., Ltd., Suwa, Japan), and analyzed by automatic image processing with WinRhizo software (version 2019a, Regent Instruments, Québec, QC, Canada). Data on total root length, root surface area, and number of root tips were used for analysis.

### 2.5. RNA Extraction and RNA-seq

For RNA extraction, the roots were ground to powder in the presence of liquid nitrogen, and RNA samples were isolated using Isogen (Nippon Gene Co., Ltd., Tokyo, Japan). Briefly, the powdered samples were homogenized in Isogen using a homogenizer (Physcotron, Microtech Co., Ltd., Funabashi, Japan), and the RNA samples were then purified using the RNeasy kit (Qiagen, Venlo, The Netherlands) and treated with RNase-free DNase (Qiagen). For each condition, three biological replicates were used. Isolation of mRNA and cDNA synthesis was carried out by using NEBNext^®^ Poly(A) mRNA Magnetic Isolation Module (NEB E7490) and NEBNext^®^ Ultra RNA Library Prep Kit for Illumina^®^ (E7530), respectively.

Next generation sequencing (NGS) analyses were conducted by Rhelixa (Tokyo, Japan). The NGS conditions were as follows: Sequence Mode: PE150 (150 bp × 2 paired-end), Instrument: Illumina NovaSeq 6000, Sofware: HiSeq Control Software 2.2.58, RTA 1.18.64, and bcl2fastq 1.8.4. The read numbers: 40.0 M reads per sample (20.0 M pairs). The raw paired-end reads were cleaned by using the Trimmomatic (ver. 0.38) program, and the cleaned reads were stored as FASTq format. The SRA accession numbers of the NGS data in GenBank/EMBL/DDBJ database are DRR450980, DRR450981, and DRR450982 for control, DRR450986, DRR450987, and DRR450988 for LPS from PNSB, and DRR450983, DRR450984, and DRR450985 for dead PNSB cells, respectively.

The rice genome IRGSP-1.0 (ver. 2021-05-10) was used for the reference sequences (number of transcript ID: 44,785), and the mapping of cleaned reads to the reference sequences was performed by HISAT2 (ver. 2.1.0). The log2 ratio of baseMean (baseMean: the mean of normalized counts of all samples) was used to calculate the fold-change in the expression of genes in two samples, and DESeq2 (1.24.0) was used to screen the differentially expressed genes (DEGs) as the criteria adjusted *p*-value (Padj) < 0.05.

### 2.6. Statistical Analysis

For statistical analyses, one-way ANOVA with a post hoc Duncan’s test was used for multiple comparison. To examine the relationship of logFC values of LPS and dead cells, Pearson’s correlation coefficient was calculated based on the distribution patterns of the two variables, and the significance of correlation coefficient was tested using the t-distribution.

## 3. Results

### 3.1. Effects of LPS from PNSB (R. sphaeroides NBRC 12203^T^) and Dead PNSB Cells on the Root Development of Rice Seedlings

To determine the concentrations of LPS and dead cells of PNSB for RNA-seq analyses, the rice seeds were aseptically germinated on agar plates containing LPS or dead cells, and grown for 7 days on vertically oriented plates. Root development of rice seedlings was assessed using WinRhizo imaging analyses. The root development was evaluated by total root length (Figure 1a,d), root surface area (Figure 1b,e), and number of root tips (Figure 1c,f). The effective concentration of LPS in root development was 10 pg/mL, and that of dead cells was 1 × 10^1^ to 1 × 10^2^ cfu/mL. Based on these results, the concentrations of LPS and dead cells for RNA-seq analyses were determined to 10 pg/mL and 1 × 10^1^ cfu/mL, respectively. The effective concentration of LPS (10 pg/mL) was further confirmed by an additional experiment (n = 5; see Appendix A for the results of the WinRhizo analysis, and Appendix A for scan images of the roots). The approximate LPS content of 1 × 10^1^ cfu/mL condition was estimated to be 5 pg/mL from the following calculation. The cell dry weight (d.w.) of PNSB culture of 1 × 10^9^ cfu/mL (stationary phase culture) is about 5 mg/mL, and the cell d.w. for 1 × 10^1^ cfu/mL culture is calculated to be 50 pg/mL. Bacterial LPS content would have been about 10% [30], and therefore the content of LPS in the cells of 1 × 10^1^ cfu/mL (50 pg d.w./mL) can be estimated to be about 5 pg/mL.

### 3.2. Effects of LPS from PNSB and Dead PNSB Cells on the Gene Expression of the Root of Rice Seedlings -RNA-Seq Analyses-

Rice seeds were germinated on agar plates containing LPS from PNSB (*R. sphaeroides* NBRC 12203^T^) at the concentration of 10 pg/mL or the dead PNSB cells at the concentration of 1 × 10^1^ cuf/mL (50 pg d.w./mL). Three days after germination, the RNA samples were extracted from the roots and analyzed by RNA-seq. Clean reads of NGS were mapped to rice genome, and the numbers of mapped genes were 34,060 for both conditions. The numbers of significantly (Padj < 0.05) up-regulated genes were 300 and 128 for LPS and dead cells, respectively, and the numbers of significantly (Padj < 0.05) down-regulated genes were 86 and 8, respectively. Figure 2 shows the Venn diagrams of the significantly up- and down-regulated genes of rice roots treated by LPS and dead cells. The total number of significantly up-regulated genes was 361, and that for down-regulated genes was 93. The number of mutual genes of significantly up-regulated genes was 67, and that for down-regulated genes was 1.

The 67 significantly up-regulated mutual genes contain 47 genes with functional annotation and 20 genes with no functional annotation (hypothetical proteins). Table 1 shows the list of these 47 genes with functional annotation sorted by logFC values of LPS. Among the 47 significantly up-regulated genes, 45 genes were stress response-related genes, clearly indicating that LPS and dead cells induced stress response in rice root. The gene of highest logFC value was thionin [31], an antimicrobial peptide against root pathogens, and there were many biotic and abiotic stress-resistant genes, such as glucan endo-1,3-beta-glucosidase [32], germin-like protein [33], Bowman–Birk type protease inhibitor [34], and late embryogenesis abundant protein [35]. The most notable genes related to biotic and abiotic stress are the genes involved in the jasmonate (JA) signaling pathway (18 genes). The list of significantly up- or down-regulated genes (361 up-regulated and 93 down-regulated genes in Figure 2) is shown in Appendix A. Among these significantly up-regulated genes, there are many genes related to tolerance against biotic (fungal and bacterial pathogens) and abiotic (cold, drought, and high salinity) stresses. There are also some genes related to ethylene and abscisic acid (ABA) signaling.

The purpose of this study is to obtain further evidence for LPS as one of the active principles of PNSB. To compare the effects of LPS and dead cells at the gene expression level, the correlation between logFC value of both conditions was examined. Figure 3a shows the scatter plot of the logFC values of significantly up-regulated genes (361 genes in Figure 2) of LPS (horizontal axis) and dead cells (vertical axis), and Figure 3b shows that (93 genes in Figure 2) of significantly down-regulated genes. In both up-regulated and down-regulated genes, the logFC values showed a significant positive relationship between LPS and dead cells with p-values (Pearson’s correlation coefficient) of 8.31 × 10^−78^ and 8.70 × 10^−13^ for up-regulated and down-regulated genes, respectively. These results indicated that most of the effects of dead cells could be accounted for by those of LPS. These results also provided supporting evidence, in terms of gene expression, for LPS as one of the active principles of PNSB. The slope of the regression line for up-regulated genes (Figure 3a) was 0.7671, while that for down-regulated genes was 0.4329. This difference in slope values between up-regulated genes and down-regulated genes indicated that the LPS was acting more potently for down-regulation than up-regulation compared to dead cells.

### 3.3. Effects of LPS from PNSB and Dead PNSB Cells on the Gene Expression of the Gene-Related JA Signaling Pathway

The jasmonates (JAs), including jasmonic acid and its derivatives, are plant hormones that control plant defenses against biotic stress, such as herbivore attack and pathogen infection [39,60]. They also play an important role in tolerance to abiotic stresses, including ozone, ultraviolet radiation, high temperatures, drought, and freezing [39,60]. As shown in Table 1, LPS and dead cells particularly stimulated the expression of genes associated with the JA signaling pathway. We examined the changes in gene expression of all genes related to the JA signaling pathway, including the genes which showed no significant change (Padj ≥ 0.05). Figure 4 shows the logFC values (vertical axis) and expression level (baseMean, horizontal axis) of 36 JA signaling pathway genes (see Appendix A for the list of these genes). The significantly (Padj < 0.05) up-regulated genes were indicated by red circles. For both LPS (Figure 4a) and dead cells (Figure 4b), a general tendency for up-regulation was observed, indicating that LPS and dead cells stimulated the JA signaling pathway. 

### 3.4. Effects of LPS from PNSB and Dead PNSB Cells on the Gene Expression of the Genes-Related Biosynthesis of Chalcone and Other Secondary Metabolites

Induction of secondary metabolites production is one of the most important reactions against herbivore attack and pathogen infection. The signaling pathways of JA, salicylic acid (SA), and ethylene are involved in inducing the production of secondary metabolites [61]. As shown in Table 1, we observed a significant up-regulation of chalcone and stilbene synthases gene; logFC value was 5.38 and 5.28 for LPS and dead cells, respectively. Chalcone synthase (CHS) and stilbene synthase (STS), are plant-specific type III polyketide synthase (PKS) [36,62]. CHS and STS catalyze a sequential decarboxylative addition of three acetate units from malonyl-CoA to a p-coumaryl-CoA starter molecule to produce naringenin chalcone and resveratrol, respectively. Naringenin chalcone is the precursor of many polyphenolic compounds, such as anthocyanins, chalcones, flavans, flavones, flavonols, and isoflavonoids. These secondary metabolites play an important role in tolerance against various biotic and abiotic stresses [63]. Figure 5 shows the logFC values and expression levels of the 21 CHS gene family (see Appendix A for the list of these genes). The significantly (Padj < 0.05) up-regulated genes were indicated by red circles. For both LPS and dead cells, a general tendency to up-regulation has been observed, indicating that the flavonoid/isoflavonoid biosynthesis pathways were up-regulated by LPS and dead cells. 

We also examined the effects of LPS and dead cells on the expression of the genes involved in other secondary metabolites biosynthesis, including terpenoid biosynthesis (30 genes) and cytochrome P450 (CYP; 233 genes), but no consistent upward or downward trend in regulation was observed in these gene groups (see Appendix A).

### 3.5. Effects of LPS from PNSB and Dead PNSB Cells on the Gene Expression of the Genes-Related Reactive Oxygen Species (ROS) Generation/Elimination

A characteristic effect of LPS in plants and animals is the generation of ROS and reactive nitrogen species (RNS) [64]. Enzymatic ROS generation is catalyzed by some oxidases such as NAD(P)H oxidase and dual oxidase, and that for RNS is catalyzed by nitric oxide synthase. In Table 1, we observed significant up-regulation of NADH oxidase genes (Os01g0370000); logFC value was 1.14 and 1.33 for LPS and dead cells, respectively, suggesting that ROS generation was accelerated by LPS and dead cells. However, there was no overall tendency for up-regulation in 14 genes coding ROS/RNS-generating enzymes (Figure 6), including one NADH oxidase, 9 NADPH oxidase, and 4 nitric oxide synthase genes (see Appendix A for the list of these genes).

To protect themselves from the toxicity of ROS, the host cells up-regulate the ROS elimination system, including SOD, catalase, and peroxidase. We examined the changes in gene expression levels related to the ROS elimination system, and found that expression of glutathione S-transferase, which has glutathione peroxidase activity, was up-regulated. Figure 7 shows the logFC values and expression levels of 57 glutathione S-transferase (GST) genes (see Appendix A for the list of these genes), and a general tendency to up-regulation has been observed. The significantly (Padj < 0.05) up-regulated genes were indicated by red circles, and 2 genes for LPS and 1 gene for dead cells were significantly up-regulated. We, however, observed no up-regulation in SOD (11 genes), catalase (5 genes), and peroxidase (ascorbate peroxidase and guaiacol peroxidase; 149 genes) (see Supplementary Appendix A).

## 4. Discussion

### 4.1. Unique Property of LPS from R. sphaeroides NBRC 12203^T^

LPS is the major component of the outer membrane of Gram-negative bacteria. LPS acts as microbe-associated molecular patterns (MAMPs) in plants as elicitors of plant innate immunity [65,66]. The receptor for LPS in plants, named lipooligosaccharide-specific reduced elicitation (LORE), has also been identified [21]. In our two previous studies [18,19], we proposed that LPS acts as one of the active principles of plant growth-promoting effects of PNSB. In the first study [18], we disrupted the PNSB cells by sonication in water, and fractionated the cell lysate by centrifugation into the supernatant and pellet, and examined their effects on the growth of *Brassica rapa* var. *perviridis* (komatsuna) through foliar feeding. We found that both the supernatant and pellet promoted the growth of the plants, and the effective concentration for supernatant and pellet were about 10^7^ cfu/mL and 10^3^ cfu/mL, respectively. The effective concentration of pellet, 10^3^ cfu/mL, was approximately one million times the dilution of the well-grown PNSB culture (10^9^ cfu/mL), and the active component in pellet was supposed to be a compound which showed biological activity at a very low concentration. Based on this finding, we expected that the LPS in the pellet could act as an active principle, and found that LPS from PNSB (*R. sphaeroides* NBRC 12203^T^) promoted the growth of the plants when added by foliar feeding at the concentrations of 10 pg/mL and 100 pg/mL. In the second study [19], we examined the effects of LPS by bio-priming method and found that bio-priming of rice seed by LPS from *R. sphaeroides* NBRC 12203^T^ at the concentration of 5 ng/mL enhanced the root growth of rice seedlings.

There have been numerous studies on the effects of LPS in plants, but compared to mammals, plants are considered to be much less sensitive to LPS [21]. The effective concentration of LPS for endotoxic reaction in mammals is generally pg/mL to ng/mL order, but that for plants was reported to be μg/mL order. Table 2 outlines 33 previous studies on the effects of LPS on plants, as well as our three studies, including this study. In the 33 previous studies, the highest concentration was 1000 μg/mL [27,67] and the lowest was 2 μg/mL [28], and in 30 studies, the LPS concentration ranged from 10 μg/mL to 100 μg/mL. The much lower effective concentration of LPS from PNSB than that of other various Gram-negative bacteria underlines the unique property of this LPS in plants. It should be also mentioned that LPS from *R. sphaeroides* NBRC 12203^T^ shows unique biological properties also found in mammals. It lacks endotoxic activity and acts as an antagonist against toxic LPSs (endotoxins) from various Gram-negative bacteria in the TLR4 signaling pathway [68,69]. This unique biological property can be attributed to its unique structure of lipid A, which is a hydrophobic domain of LPS that anchors LPS to the outer membrane of Gram-negative bacteria. The chemically synthesized lipid A of *R. sphaeroides*, named eritoran (E5564; Eisai), which protects mammals from a lethal cytokine storm by blocking the TLR4 signaling pathway [70], has also been developed for therapeutic application [71]. 

### 4.2. Comparison of the Effects of LPS and Dead Cells of PNSB on the Gene Expression of Rice Seedlings

In the present study, to gain a better understanding of the effects of LPS from PNSB (*R. sphaeroides* NBRC 12203^T^) in plants, we examined the effects of LPS on the gene expression of roots of rice seedlings by RNA-seq analyses. We first compared the effects of LPS and dead cells of PNSB on the gene expression of rice seedlings. In the scatter plots of logFC values of significantly up- and down-regulated genes of LPS and dead cells (Figure 2), there was a significant positive relationship between LPS and dead cells. This result indicated that most of the effects of dead cells could be explained by those of LPS, and provided evidence in support of the LPS as one of the active principles of PNSB at the gene expression level. The difference in the slope values in the scatter plots (Figure 2) for up-regulation (0.7671) and down-regulation (0.4329) indicated that the effects of LPS were more potent for down-regulation than up-regulation, and this suggests that other cellular components act differently with LPS for up- and down-regulation.

### 4.3. Effects of LPS on the Expression of Genes Related to the JA Signaling Pathway

As shown in Table 1, the LPS from PNSB obviously stimulated the expression of genes related to the JA signaling pathway. The innate immune response in plants is regulated through multiple signaling networks, including SA, JA, and ethylene [95]. JA is a plant hormone that is inductively synthesized in response to wounding, pathogen infections, and attack by herbivorous arthropods. The plant pathogens, which stimulate the JA signaling pathway, are fungi, such as *Alternaria brassiicola*, *Pythium irregulare*, *Botrytis cinerea*, and *Magnaporthe oryzae*, and bacteria such as *Pseudomonas syringae* [60].

Several studies in Table 2 indicated the involvement of the JA signaling pathway in the response of plants to LPS. Finnegan et al. [25] reported that in *Arabidopsis thaliana* there was an induction of phytoalexin synthesis by the LPS from *Burkholderia cepacian*. Their metabolome analysis revealed an increase in JA and jasmonoyl-L-isoleucine, indicating the involvement of a JA signaling pathway. Tinte et al. [81] reported in *A. thaliana* the induction of biosynthesis of defense-related metabolites including 9,10-dihydrohydroxy-jasmonic acid sulphate, a catabolite of the JA signaling pathway. Mareya et al. [93] examined the induction of secondary metabolites synthesis by LPS from *Burkholderia andropogonis* in *Sorghum bicolor* using a metabolome analysis, and revealed that the JA and methyl jasmonic acid (MeJA) accumulated to significantly high levels in LPS-treated plants. Shilina et al. [80] examined the effects of combined treatment with LPS from pathogenic *Pseudomonas aeruginosa* strain 9096 and SA on the disease resistance of wild-type (Col-0) and *jin1* mutant (with impaired JA signaling) *A. thaliana*. They found that treatment of *A. thaliana* seeds with a composite preparation (LPS and SA) increased the resistance of seedlings to *P. aeruginosa*, and the protective effect was more pronounced in *jin1* mutant, indicating the possibility of compensation for JA signaling impairment by activation of the SA signaling pathway. Rapicavoli et al. [94] reported that in *Vitis vinifera,* the LPS from *Xylella fastidiosa* induced the expression of genes of secondary metabolite biosynthesis, such as chalcone/stilbene synthase and chalcone reductase, as observed in the current study. Their GO enrichment analysis on RNA-seq data indicated the involvement of SA and JA signaling pathways in the elicitation of defense response by LPS.

In Table 2, several studies also indicated the involvement of an SA signaling pathway in the response of plants to LPS. Mishina and Zeier [73] examined the induction of systemic acquired resistance (SAR) by LPS from *P. syringae* and *E. coli* in *A. thaliana*. The LPS treatment elevated the level of SA, suggesting that LPS affected the SA signaling pathway. They also used a JA-insensitive *A. thaliana jar1* mutant, and found that the effect of LPS was independent of the JA signaling pathway. Stomatal closure is one of the plant’s defense responses to pathogen infection, and this response is regulated by ABA signaling. Melotto et al. [78] examined the stomatal closure in *A. thaliana* in response to LPS from *P. aeruginosa* by using a cronatine-insensitive mutant *coi1-20*, and indicated the response to LPS was regulated through both ABA and SA signaling. Iizasa et al. [23] reported the induction of defense response by LPS from *P. aeruginosa* in *A. thaliana.* The gene ontology (GO) analysis of RNA-seq revealed that GO terms, “response to bacterium”, “response to SA stimulus”, and “response to abscisic acid (ABA) stimulus” were enriched in up-regulated genes, suggesting the involvement of SA and ABA signaling in the response of plants to LPS. Sanabria et al. [90] reported that LPS from *B. cepacia* induced in *Nicotiana tabacum* the defense response S-domain receptor-like kinase, which plays a crucial role in sensing the presence of invading micro-organisms, and suggested the involvement of the SA pathway in this reaction.

### 4.4. Effects of LPS on the Expression of Genes Related to Secondary Metabolism

Production of secondary metabolites, such as phenylpropanoids, terpenoids, polyketides, and alkaloids, is one of the most important defense mechanisms of plants against microbial pathogens and also herbivores [96]. As shown in Figure 5, we observed significant up-regulation of the CHS gene family. CHS is a key enzyme of the flavonoid/isoflavonoid biosynthesis pathway [36]. A CHS plays a crucial role in verticillium disease resistance in plants, and Lei et al. [97] reported that knockdown of the CHS gene in both *A. thaliana* and cotton (*Gossypium arboreum*) increased their susceptibility to *V. dahliae* infection. In Table 2, several studies reported the up-regulation of secondary metabolism by LPS. Rapicavoli et al. [94] reported the induction of a defense response against pathogens in grapevine (*Vitis vinifera*) by LPS from *Xylella fastidiosa*. They found an increase in the expression of enzymes contributing to phenylpropanoid biosynthesis, such as a chalcone/stilbene synthase, as we observed in the present study, and a chalcone reductase. Finnegan et al. [25] examined the changes in secondary metabolism in *A. thaliana* elicited by LPS from *B. cepacia* by liquid chromatography coupled to mass spectrometry detection (LC-MS) analysis and found the induction of biosynthesis of phytoalexin, including various flavonoids. Tinte et al. [81] compared the effects of LPS from *B. cepacian*, *P. syringae*, and *X. campestris* on the defense-related metabolites synthesis, and found that secondary metabolites, such as flavonoids and terpenoids, were differentially up-regulated by the different kinds of LPS. Mhlongo et al. [91] reported the induction of phenylpropanoid biosynthesis in the cell culture of *Nicotiana tabacum* elicited by LPS from *B. cepaci* by metabolome analysis with LC-MS. They also showed that the treatment with the phytohormones SA, MeJA, and ABA resulted in differentially induced phenylpropanoid pathway metabolites. Newman et al. [92] reported the induction of the phenylpropanoid pathway and of tyramine metabolism by LPS from *X. campestris* pv. *campestris* in leaves of pepper (*Capsicum annuum*). They observed the accumulation of SA, coumaroyl-tyramine (CT), and feruloyl-tyramine, and also up-regulation of the genes of tyramine hydroxycinnamoyl transferase (THT) and phenylalanine ammonia lyase (PAL) by LPS treatment. Mareya et al. [93] used cell suspension culture of *Sorghum bicolor* and examined the effects of LPS from *B. andropogonis* using untargeted metabolomics with LC-MS, and found the accumulation of defense-related metabolites, including phenylpropanoids, indole alkaloids, and oxylipins. In the present study, we observed no consistent up-reregulation of the genes related to terpenoid biosynthesis (30 genes) and CYP genes (233 genes) (see Appendix A). However, we anticipate that these secondary metabolisms will also be up-regulated at some point after exposure to PNSB, because various secondary metabolisms are involved in the defense reaction of plants. The results of the RNA-seq analysis of the present study only reflect the effects of LPS at a specific time (3 days after germination), and further experiments on the time course of LPS effects, are needed to evaluate the effects of LPS from PNSB on the overall secondary metabolisms in plants.

### 4.5. Effects of LPS on the Expression of Genes Related to ROS Generation/Elimination

Generating ROS is a typical effect of LPS in plants [98,99], and many studies listed in Table 2 reported on ROS generation by LPS. In the present study, we observed significant up-regulation of the gene of NADH oxidase (Os01g0370000), a ROS-generating enzyme, but there was no consistent tendency of up-regulation of the genes of 14 ROS-generating enzymes (Figure 5). With respect to ROS elimination, 2 GST genes (Os01g0371500 and Os10g0530900) were significantly upregulated by LPS, and a general trend was observed in the up-regulation of the genes of GST isozymes (Figure 6). Similar to our results, Van Loon et al. [99] reported the induction of ROS generation and up-regulation of GST in tobacco cell suspension culture elicited by the LPS-containing cell walls of *Pseudomonas putida* WCS358.

### 4.6. Promotion of Root Development by LPS

As regards promoting root development by LPS (Figure 1), a possible explanation of its mechanism is that a generation of weak ROS (non-harmful level) by LPS promoted root development. It is well established that a weak ROS stimulated root development [100,101]. Three studies listed in Table 2 reported contradictory results on the effects of ROS on plant development, two promotive and one inhibitory. Chávez-Herrera et al. [28] examined the effects of LPS from *Azospirillum brasilense* Sp245 on the leaf and root development of *Triticum aestivum* by pot experiment, and found that both leaf and root development were promoted by LPS. The same research group reported the ROS (H_2_O_2_) generation and plant growth promotion (leaf and root) by LPS from *A. brasilense* Sp245 in *T. aestivum* [85]. They also treated the LPS-elicited plants with catalase, which catalyzes the decomposition of hydrogen peroxide to water and oxygen, and found a decrease in H_2_O_2_ content in the roots and also a decrease in the growth of seedlings. Their results, therefore, indicated that the generation of ROS by LPS promoted the growth of plants. In contrast to their conclusions, Shang-Guan et al. [24] reported that the generation of ROS by LPS from *P. aeruginosa* inhibited seeding growth in *A. thaliana*. To clarify the mechanisms for promoting root development by LPS from *R. sphaeroides*, more detailed studies on the relationship between ROS generation and root development in LPS-treated plants, including the experiments using inhibitors of LPS, such as polymyxin B, and that of NAD(P)H oxidase, such as diphenyleneiodonium chloride, and detection of ROS by image analyzing, are required, and these studies are being carried out in our laboratory.

### 4.7. Stimulation of Defense Response by LPS

This study is the first report that showed PNSB stimulate the defense response of plants at the level of gene expression. According to the review article by Sakarika et al. [3], there have been 10 studies on the enhanced tolerance against abiotic stress by PNSB treatment. The mechanisms which underlie these effects have been discussed mainly with respect to the effects of ALA produced by PNSB, because ALA is a precursor of the synthesis of chlorophyll, vitamin B12, anti-oxidative enzymes, and other metabolites which ameliorate various abiotic stresses [16,102]. The results obtained in this study, therefore, shed new light on the mechanisms behind the protective effects of PNSB against abiotic stress in plants. To our knowledge, there has been only one study on the enhanced tolerance against biotic stress by PNSB. Su et al. [103] reported an enhanced resistance of plants, by PNSB treatment, against tobacco mosaic virus (TMV) infection. One possible explanation for the small number of examples on the protective effects of PNSB against biotic stress is an overdose of PNSB cells. As shown in Figure 1, an adequate concentration for LPS was about 10 pg/mL and that for PNSB dead cells was about 10^1^ to 10^2^ cfu/mL, and higher concentrations led to lower effects. PNSB concentrations in most studies presented in the review article by Sakarika et al. [3] are 10^7^ to 10^8^ cfu/mL and/or above 10^9^ cfu/plants. The results of the present study therefore suggest the need to re-examine the optimal concentration of PNSB in plant treatment.

### 4.8. Practical Application of LPS

From a practical point of view, the much lower effective concentration of LPS from PNSB compared with LPS from other bacteria is a great advantage for application in agriculture. In addition to plant growth promotion, the up-regulation of various genes related to biotic and abiotic stresses as shown in Table 1 and Appendix A indicated a potential of LPS and dead cells of PNSB for stimulation of a defense response in plants and enhance their tolerance both against biotic and abiotic stresses.

Another potential application of LPS from PNSB is the stimulation of secondary metabolisms in plant cell cultures. Plant secondary metabolites are considered important sources of pharmaceuticals, food additives, flavors, cosmetics, and other industrial products. The JA regulates the synthesis of secondary metabolites in a wide range of plant species [104,105], and there has been a lot of research on the enhancement of secondary metabolites production of plant cell cultures by exogenous application of JA and MeJA [106,107]. Chen et al. [108] also reported that constitutive activation of the JA signaling pathway by genetic engineering in tomato enhanced the production of secondary metabolites. Thus the application of LPS from PNSB and also with a combination of JA/MeJA to plant cell culture must be a promising approach to improve secondary metabolite productivity in plant cell culture.

Finally, another matter of interest is the question of the effects of lipid A of LPS from *R. sphaeroides*. Among the studies listed in Table 2, seven [24,65,66,72,75,76,84] examined the effects of lipid A, a hydrophobic domain of LPS, in comparison with those of whole LPS, and found that lipid A also induced defense response. It is therefore of a great interest to examine whether the lipid A of *R. sphaeroides* and also eritoran (E5564; Eisai), the chemically synthesized analog of lipid A of *R. sphaeroides,* can induce a defense response in plants like as LPS of *R. sphaeroides* does. Eritoran is an endotoxin-antagonist for the TLR4 signaling pathway, and has been developed for therapeutic application to suppress a cytokine storm in patients [70,71]. Particularly, if eritoran can stimulate a defense response in plants, it is quite advantageous for agricultural use, because compared to the natural lipid A of *R. sphaeroides*, we can expect higher uniformity of material quality of chemically synthesized eritoran. Eritoran has undergone clinical trials and its cost-effective manufacturing process is also beneficial for practical application.

## 5. Conclusions

The results obtained with rice (*Oryza sativa*) seedlings provided supporting evidence, in terms of gene expression, to our previous proposal that LPS acts as one of the active principles of PNSB in plants. This study is the first report that showed PNSB stimulate the defense response of plants at the level of gene expression, and up-regulated genes including the genes of the JA signaling pathway and various genes related to biotic and abiotic stress tolerances, despite the non-pathogenicity of PNSB. A significant up-regulation of CHS, a key enzyme of the flavonoid/isoflavonoid biosynthesis pathway, gene family by LPS was also observed. LPS from *R. sphaeroides* stimulated the gene expression at the concentration of 10 pg/mL, and this concentration was a thousand to a million times lower compared to those reported in previous studies on other LPS from various Gram-negative bacteria. From a practical point of view, this much lower effective concentration of LPS from PNSB is a great advantage for application in agriculture, and also for stimulation of secondary metabolisms in plant cell culture.

## Figures and Tables

**Figure 1 microorganisms-11-01676-f001:**
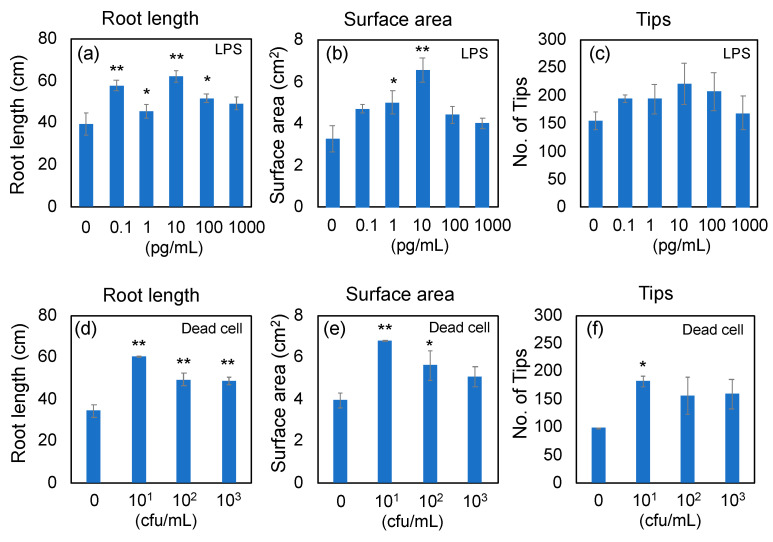
Effects of LPS and dead cells of *Rhodobacter sphaeroides* NBRC 12203^T^ on root development of rice seedlings. Rice seeds were aseptically germinated on agar plates, and grown for 7 days on vertically oriented plates containing LPS and dead cells of *Rhodobacter sphaeroides* NBRC 12203^T^. Root development was evaluated by WinRhizo imaging analyses. The root development was evaluated by total root length (**a**,**d**), root surface area (**b**,**e**), and number of root tips (**c**,**f**). Data are means ± SD (n = 3). Asterisk indicates the result that was significantly different (* *p* < 0.05, ** *p* < 0.01) from the control.

**Figure 2 microorganisms-11-01676-f002:**
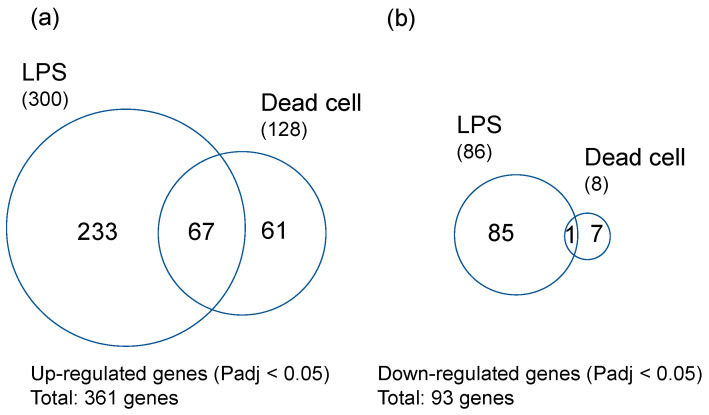
Venn diagrams of the significantly up- and down-regulated genes of rice roots treated by LPS and dead cells: (**a**) Significantly (Padj < 0.05) up-regulated genes (total gene number: 361) and (**b**) significantly (Padj < 0.05) down-regulated genes (total gene number: 93). RNA-seq analysis was carried out with three biological replicates.

**Figure 3 microorganisms-11-01676-f003:**
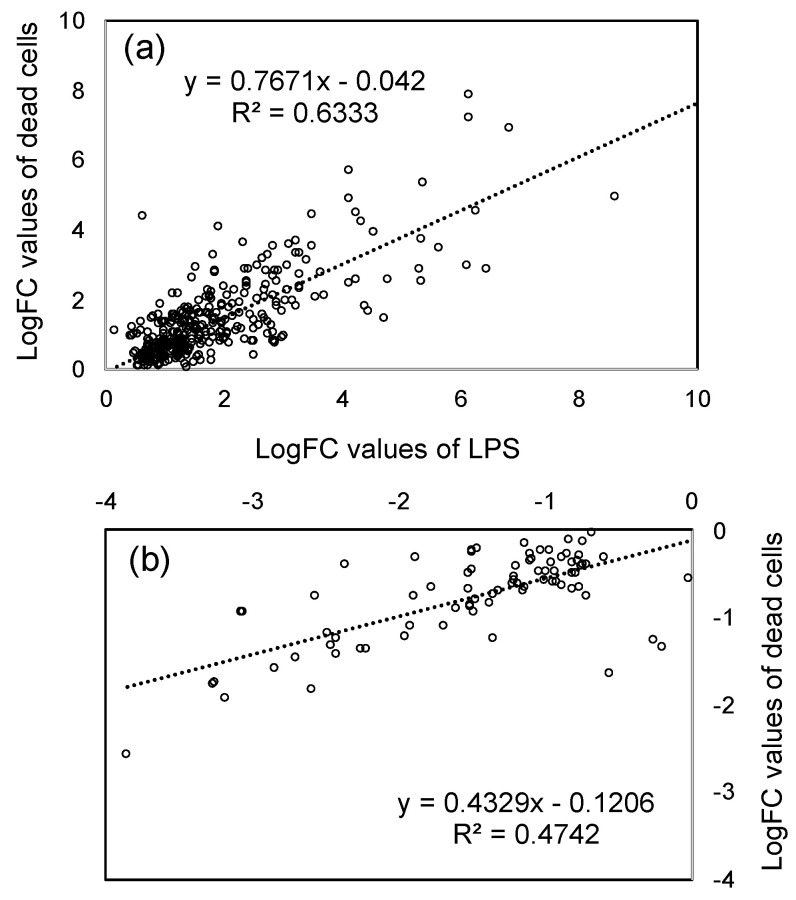
Scatter plot of the logFC values of significantly up- and down-regulated genes of LPS and dead cells: (**a**) Significantly up-regulated genes (361 genes in Figure 2a), and (**b**) significantly down-regulated genes (93 genes in Figure 2b). Horizontal axis: LPS; vertical axis: dead cells.

**Figure 4 microorganisms-11-01676-f004:**
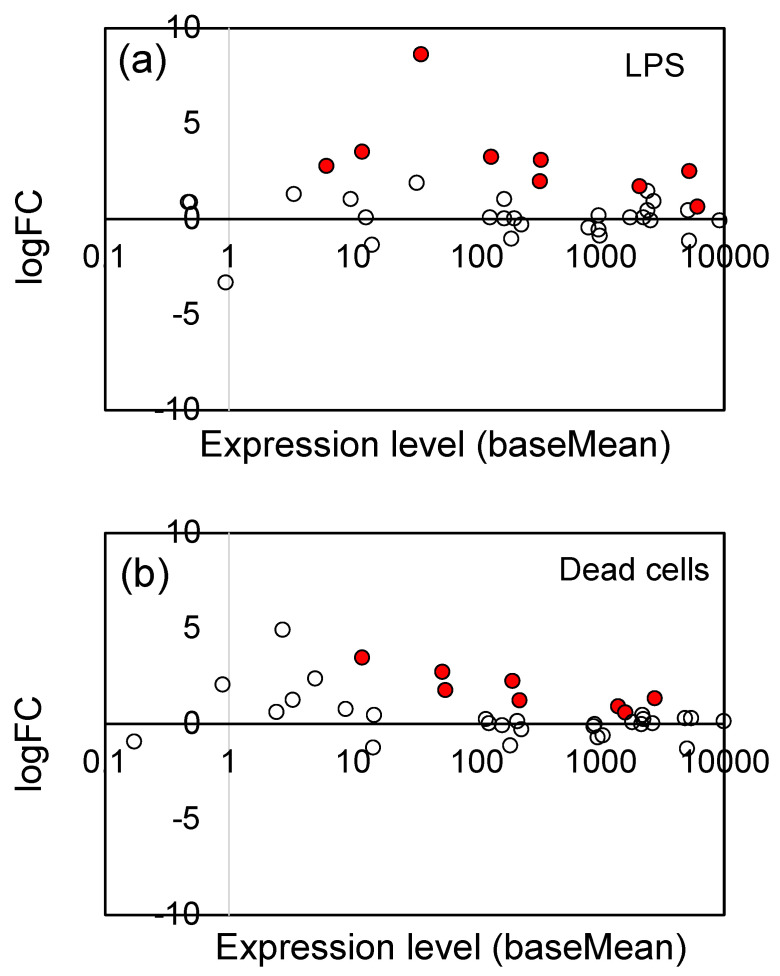
Plot of the logFC values and expression levels (baseMean) of 36 JA signaling pathway genes: (**a**) LPS and (**b**) dead cells. The significantly (Padj < 0.05) up-regulated genes were indicated by red circles (●), and the open (white) circles are the genes which showed no significant change. See Appendix A for the list of 36 genes.

**Figure 5 microorganisms-11-01676-f005:**
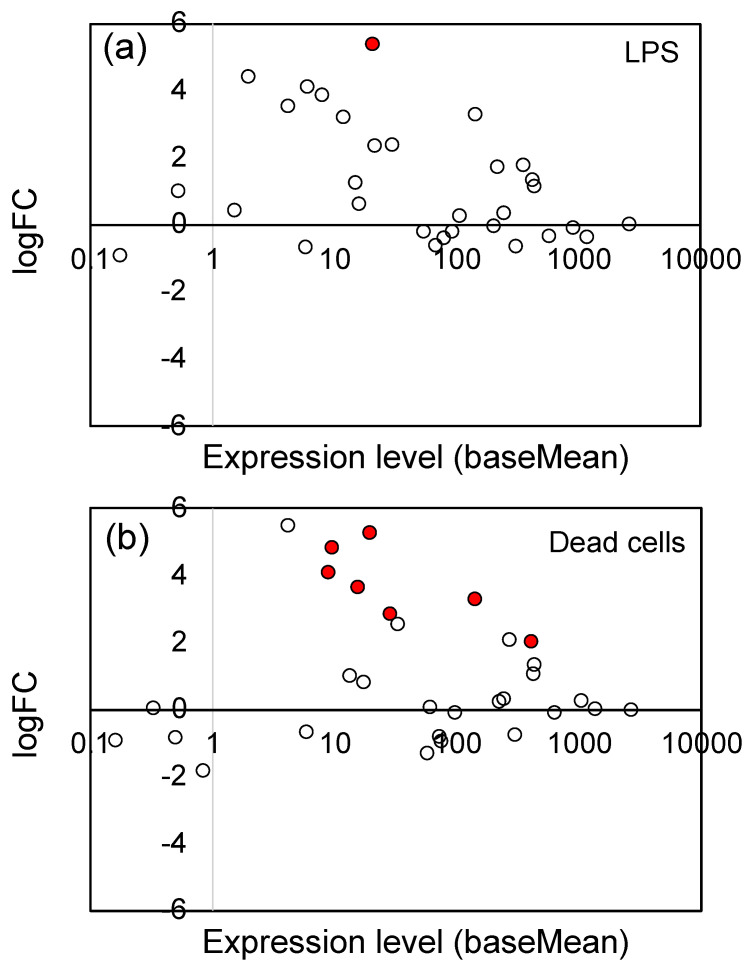
Plot of the logFC values and expression levels (baseMean) of the 21 chalcone synthase gene family: (**a**) LPS and (**b**) dead cells. The significantly (Padj < 0.05) up-regulated genes were indicated by red circles (●), and the open (white) circles are the genes which showed no significant change. See Appendix A for the list of 21 genes.

**Figure 6 microorganisms-11-01676-f006:**
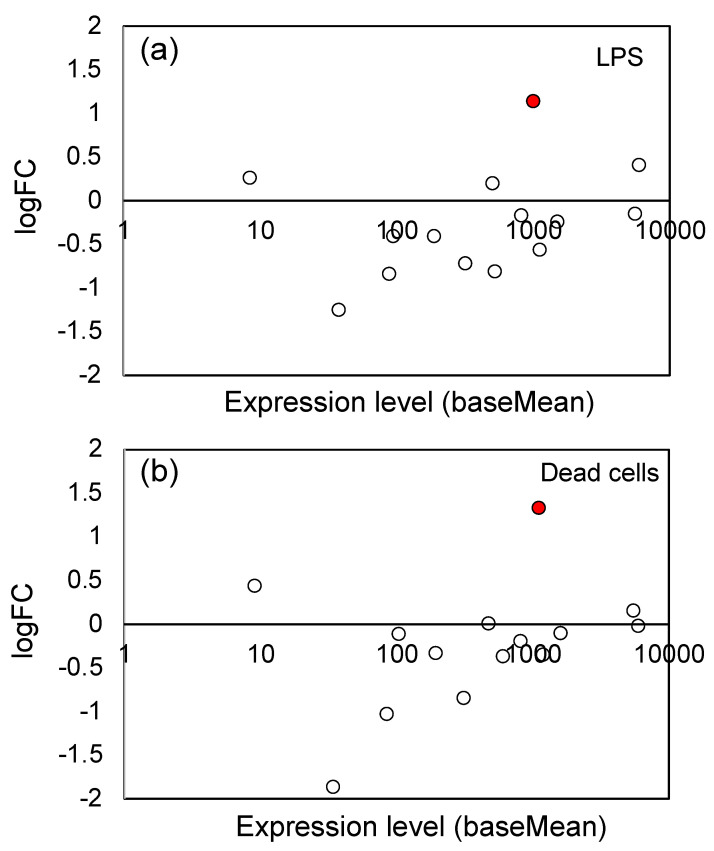
Plot of the logFC values and expression levels (baseMean) of 14 genes coding ROS/RNS-generating enzymes: (**a**) LPS and (**b**) dead cells. The significantly (Padj < 0.05) up-regulated genes were indicated by red circles (●), and the open (white) circles are the genes which showed no significant change. See Appendix A for the list of 14 genes.

**Figure 7 microorganisms-11-01676-f007:**
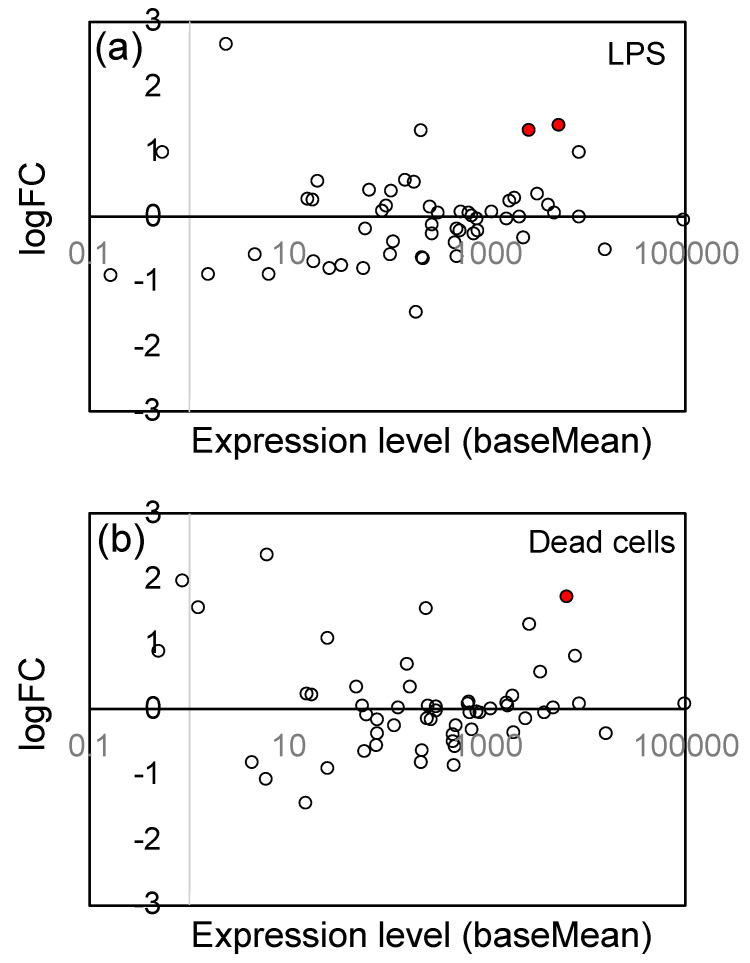
Plot of the logFC values and expression levels (baseMean) of 57 glutathione S-transferase (GST) genes: (**a**) LPS and (**b**) dead cells. The significantly (Padj < 0.05) up-regulated genes were indicated by red circles (●), and the open (white) circles are the genes which showed no significant change. See Appendix A for the list of 57 genes.

**Table 1 microorganisms-11-01676-t001:** List of 47 significantly (Padj < 0.05) up-regulated genes, in both conditions, with functional annotation sorted by logFC values of LPS.

	LPS	Dead Cells				
Gene ID	logFC		Gene Description	Stress Response	JA *	Reference
Os06g0514100	6.17	7.82	Thionin; antimicrobial peptide	X		[31]
Os10g0177300	5.38	5.28	Chalcone and stilbene synthases	X		[36]
Os03g0183500	4.54	3.88	FCS-like zinc finger protein	X	X	[37,38]
Os11g0603000	4.34	4.17	Basic helix-loop-helix dimerization region bHLH domain containing protein	X	X	[38,39]
Os02g0137700	4.25	2.53	NAD(P)-binding domain containing protein	X		[40]
Os01g0946600	3.65	2.74	Glucan endo-1,3-beta-glucosidase	X		[32]
Os10g0392400	3.55	2.01	Tify domain containing protein	X	X	[38,39]
Os07g0153000	3.51	3.48	Jasmonate ZIM-domain (JAZ) family	X	X	[38,39]
Os06g0513781	3.41	3.09	Thionin; antimicrobial peptide	X		[31]
Os03g0741100	3.29	2.67	Basic helix-loop-helix transcription factor, drought tolerance	X	X	[38,39]
Os11g0684000	3.25	1.76	Myb transcription factor	X	X	[38,39]
Os01g0946700	3.17	1.93	Glucan endo-1,3-beta-glucosidase	X		[32]
Os03g0142600	3.10	3.52	Myb transcription factor	X	X	[38,39]
Os03g0180900	3.09	2.22	Jasmonate ZIM-domain containing protein	X	X	[38,39]
Os10g0118200	3.08	2.94	Acetylserotonin O-methyltransferase	X		[41]
Os01g0124700	3.04	1.93	Bowman–Birk type protease inhibitor	X		[34]
Os03g0181100	2.92	2.78	Tify domain containing protein	X	X	[38,39]
Os12g0478400	2.90	2.46	EGF-type aspartate/asparagine hydroxylation site domain containing protein			
Os01g0225500	2.85	3.47	3-methyl-2-oxobutanoate hydroxymethyltransferase	X		[42]
Os01g0124650	2.91	1.84	Bowman–Birk type protease inhibitor	X		[34]
Os05g0161500	2.73	2.75	Calcium-activated (p)ppGpp synthetase	X		[43]
Os02g0808000	2.57	1.98	Wall-associated receptor kinase 2	X		[44]
Os05g0586200	2.53	1.35	Gretchen hagen 3 (GH3)	X		[45]
Os08g0190100	2.38	2.12	Germin-like protein	X		[33]
Os12g0503000	2.08	2.08	Allantoin transporter	X		[46]
Os09g0439200	1.97	1.24	Jasmonate ZIM-domain protein	X	X	[38,39]
Os01g0108600	1.76	1.51	Basic helix-loop-helix dimerization region bHLH domain containing protein	X	X	[38,39]
Os02g0181300	1.75	1.56	WRKY transcription factor	X	X	[38,47]
Os03g0402800	1.71	0.89	TIFY family protein, JASMONATE-ZIM domain (JAZ) protein	X	X	[38,39]
Os03g0198600	1.71	1.56	Homeodomain-leucine zipper transcription factor; regulation of panicle exertion	X	X	[38,48]
Os12g0138800	1.59	1.92	Six-bladed beta-propeller, TolB-like domain containing protein	X	X	[38,49]
Os12g0548401	1.58	1.64	Proteinase inhibitor	X		[50]
Os06g0231600	1.53	1.13	RING-H2 finger protein ATL1Q	X		[51]
Os07g0475900	1.50	1.52	ACT domain containing protein kinase	X		[52]
Os12g0267200	1.46	1.32	Cyclopropane-fatty-acyl-phospholipid synthase	X		[53]
Os09g0401300	1.45	0.57	JASMONATE ZIM-domain (JAZ)	X	X	[38,39]
Os07g0138200	1.37	1.51	NAC transcription factor; ABA-induced leaf senescence and tillering	X		[54]
Os01g0314800	1.32	1.11	Late embryogenesis abundant protein 3	X		[35]
Os11g0644700	1.29	1.61	Plant disease-resistance response protein	X		[55]
Os06g0112100	1.17	1.03	Nucleoside phosphorylase			
Os05g0126800	1.15	1.02	Mss4-like domain containing protein	X		[56]
Os01g0370000	1.14	1.33	NADH:flavin oxidoreductase/NADH oxidase	X		[57]
Os05g0542200	1.06	0.69	Alpha/beta hydrolase fold-1 domain containing protein	X		[58]
Os09g0248900	1.04	1.30	Myb/SANT-like domain containing protein	X	X	[38,39]
Os07g0633400	0.90	1.51	IQ calmodulin-binding region domain containing protein	X		[59]
Os12g0626400	0.82	0.84	Phytoene synthase 1			
Os04g0517100	0.74	0.61	SG2-type MYB transcription factor; cold tolerance; resistance to fungal and bacterial pathogens	X	X	[38,39]

* Genes involved in jasmonate (JA) signaling pathway.

**Table 2 microorganisms-11-01676-t002:** Outline of previous studies on the effects of LPS in plants.

Plant	Bacteria	Concentration (μg/mL)	Response to LPS	Reference
*Arabidopsis thaliana*	*Burkholderia cepacia*	100	Activation of nitric oxide synthase (NOS) and induction of defense genes	[72]
*Arabidopsis thaliana*	*Xanthomonas campestris*	50	Elicitation of innate immunity	[65]
*Arabidopsis thaliana*	*Pseudomonas syringae* *E. coli*	100	Induction of systemic acquired resistance (SAR)	[73]
*Arabidopsis thaliana*	*Pseudomonas chlororaphis* O6	100	Stomatal closure and induction of systemic tolerance to drought	[74]
*Arabidopsis thaliana*	*Xanthomonas campestris*	50 (lipid A)	Induction of pathogenesis-related 1 (PR1) gene	[75]
*Arabidopsis thaliana*	*Burkholderia cepacia*	20 (lipid A).	Induction of defense-related genes	[76]
*Arabidopsis thaliana*	*Burkholderia cepacia*	100 (LPS) 20 (lipid A)	Induction of defense-related metabolites synthesis	[66]
*Arabidopsis thaliana*	*Pectobacterium atrosepticum* *Pectobacterium carotovorum* subsp. *carotovorum*	10 to 100	Induction of defense response	[77]
*Arabidopsis thaliana*	*Burkholderia cepacia*	80	Induction of phytoalexin synthesis	[25]
*Arabidopsis thaliana*	*Pseudomonas aeruginosa*	100	Stomatal closure	[78]
*Arabidopsis thaliana*	*E. coli* *Pseudomonas aeruginosa*	LPS (*E.*) 10 LPS (*P.*) 100	ROS generation	[23]
*Arabidopsis thaliana*	*Pseudomonas aeruginosa*	100	Induction of defense response	[79]
*Arabidopsis thaliana*	*Pseudomonas aeruginosa*	LPS 25 Lipid A 10	ROS generation Inhibition of seedling growth	[24]
*Arabidopsis thaliana*	*Pseudomonas aeruginosa*	100	Enhanced resistance to pathogen Activation of SA signaling pathway	[80]
*Arabidopsis thaliana*	*Burkholderia cepacian* *Pseudomonas syringae* *Xanthomonas campestris*	100	Induction of defense-related metabolites synthesis	[81]
*Arabidopsis thaliana*	*Xanthomonas campestris*	100	Induction of defense response	[82]
*Oryza sativa*	*Xanthomonas oryzae*	50	ROS generation	[22]
*Oryza sativa*	*Xanthomonas oryzae*	100	Induction of defense response	[83]
*Oryza sativa*	*Pseudomonas aeruginosa, E. coli*	50	Induction of immune response	[84]
*Solanum tuberosum*	*Rhizobium etli* strain G12	100 to 1000	Resistance to nematode infection	[27]
*Triticum aestivum*	*Azospirillum brasilense* Sp245	100	Growth promotion, ROS generation	[85]
*Triticum aestivum*	*Azospirillum brasilense* Sp245	2–5	Promotion of plant development (plant aging, spike formation, and size)	[28]
*Triticum aestivum*	*Azospirillum brasilense* SR8	1000	Root hair deformations	[67]
*Nicotiana tabacum*	*Burkholderia cepacia*	100	Induction of defense response	[86]
*Nicotiana tabacum*	*Burkholderia cepacia*	100	ROS generation	[87]
*Nicotiana tabacum*	*Xanthomonas campestris*	10 (5 to 500)	ROS generation	[64]
*Nicotiana tabacum*	*Xanthomonas campestris* pv. *campestris*	20	ROS generation	[88]
*Nicotiana tabacum*	*Burkholderia cepacia*	100	Induction of innate immunity ROS generation	[89]
*Nicotiana tabacum*	*Burkholderia cepacia*	100	Induction of defense response S-domain receptor-like kinase (RLK)	[90]
*Nicotiana tabacum*	*Burkholderia cepacia*	100	Induction of phenylpropanoid biosynthesis	[91]
*Capsicum annuum*	*Xanthomonas axonopodis* pv. *Vesicatoria* *X. campestris* pv. *campestris*	50	Accumulation of salicylic acid (SA), coumaroyl-tyramine (CT), and feruloyl-tyramine (FT)	[92]
*Sorghum bicolor*	*Burkholderia andropogonis*	100	Induction of secondary metabolites synthesis	[93]
*Vitis vinifera*	*Xylella fastidiosa*	50	Induction of defense response against pathogens	[94]
	PNSB	Concentration		
*Brassica rapa* var. *perviridis*	*Rhodobacter sphaeroides* NBRC 12203^T^	10 pg/mL	Growth promotion	[18]
*Oryza sativa*	*R. sphaeroides* NBRC 12203^T^	5 ng/mL	Promotion of root development	[19]
*Oryza sativa*	*R. sphaeroides* NBRC 12203^T^	10 pg/mL	Stimulation of JA signaling pathway Induction of secondary metabolites synthesis	Present study

## Data Availability

The data presented in this study are available upon request from the corresponding author.

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
