# Peer review of "Effects of LPS from Rhodobacter sphaeroides, a Purple Non-Sulfur Bacterium (PNSB), on the Gene Expression of Rice Root"

_microorganisms, 2023, doi:10.3390/microorganisms11071676_

Round 1

Reviewer 1 Report

1. As Figure 1, the author presents the experimental results from only 3 samples (n = 3), which is too small. The authors should increase the number of samples used in the study to obtain accurate results.

2. The authors should add photographs of the rice plants to show the differences in the results as to which concentrations affect growth.

3. The authors report that LPS affects the up-regulated genes involved in the jasmonate signaling pathway. In addition to RNA-seq analysis, the authors should reconfirm the results by qPCR and investigate whether protein expression is increased following gene expression was increased.

Author Response

Dear Reviewer,

Thank you very much for your time and effort to review our manuscript, and your quite helpful comments and suggestions to improve our manuscript.

(Comment)

  1. As Figure 1, the author presents the experimental results from only 3 samples (n = 3), which is too small. The authors should increase the number of samples used in the study to obtain accurate results. 

(Answer)

We agree with reviewer’s comment that the number samples (n = 3) is too small. To our regret, however, we don’t have the results of experiments of the same design to Figure 1 with a larger sample number. Instead, we would like to present the results of another experiment to confirm the effects of LPS at the concentration of 10 pg/mL with 5 biological replicates (n = 5) as supplementary Figure S1 in the revised manuscript. Please note the experimental method for Figure 1 and Figure S1 was different. We used agar plates for Figure 1, and 15 mL test tubes for Figure S1. This is because we observed smaller seasonal variation, due to the aging of seeds, with the test tube method.

(Comment)

  1. The authors should add photographs of the rice plants to show the differences in the results as to which concentrations affect growth.

(Answer)

As suggested by the reviewer, we showed the scan image of the above mentioned experiments (n = 5) as Figure S1(b).

(Comment)

  1. The authors report that LPS affects the up-regulated genes involved in the jasmonate signaling pathway. In addition to RNA-seq analysis, the authors should reconfirm the results by qPCR and investigate whether protein expression is increased following gene expression was increased.

(Answer)

We totally agree with reviewer’s suggestion that we should reconfirm the results by qPCR. We, however, would like to do qPCR experiments in future research for the following reasons.

  • The main aim of this research is to show a supportive evidence for our proposal that LPS acts as one of the active principles of PNSB at the gene expression level, and we believe RNA seq data is sufficient to support our proposal.
  • The results of the present study suggested the necessity for time course studies of the effects of LPS, because the results of RNA-seq analysis of the present study only reflect the effects of LPS at a specific time (3 days after germination, please see lines 443 to 446 of the revised manuscript). We, therefore, want to perform detailed qPCR analyses, including time course studies, as a separate study.

For protein expression, we think the analyses of protein expression require a lot of experimental works, and not necessarily required for the publication of the result of RNA-seq analysis, but we will be trying to investigate whether protein expression is increased following gene expression was increased in the future.

Given this comment (comment No. 3) of the reviewer, we noticed that the title of this article “Stimulation of jasmonate signaling pathway by lipopolysac-charide (LPS) from Rhodobacter sphaeroides, a purple non-sulfur bacterium (PNSB), in rice root” is inappropriate because we have not confirmed the up-regulation of JA signaling as suggested by the reviewer, and we changed the title of this article to “Effects of LPS from Rhodobacter sphaeroides, a purple non-sulfur bacterium (PNSB), on the gene expression of rice root”.

Thank you again.

Reviewer 2 Report

The present manuscript investigates the effects of lipopolysaccharide from R. sphaeroides (effective concentration of 10 pg/ml) on the gene expression of the root rice. The research is interesting and is well presented and organised. Some minor comments, lines 32-40 contain useful information but there is no reference. The aim of the study could be more analytically presented, as well as the section of the conclusions, which could be more comprenhensive. Line 151 check spelling. 

Moderate editing of English language and minor spelling mistakes

Author Response

Dear Reviewer,

Thank you very much for your time and effort to review our manuscript, and your quite helpful comments and suggestions to improve our manuscript.

(Comment)

The present manuscript investigates the effects of lipopolysaccharide from R. sphaeroides (effective concentration of 10 pg/ml) on the gene expression of the root rice. The research is interesting and is well presented and organised. Some minor comments, lines 32-40 contain useful information but there is no reference. The aim of the study could be more analytically presented, as well as the section of the conclusions, which could be more comprenhensive. Line 151 check spelling.

(Answer)

> lines 32-40 contain useful information but there is no reference.

As suggested by the reviewer, we added following two references which comprehensively describe the characteristic of membranes of photosynthetic bacteria.

  1. Oelze, J.; Drews, G. Membranes of photosynthetic bacteria. Biochim Biophys Acta - Rev Biomembr 1972, 265, 209–239, doi:https://doi.org/10.1016/0304-4157(72)90003-2.
  2. George, D.M.; Vincent, A.S.; Mackey, H.R. An overview of anoxygenic phototrophic bacteria and their applications in environmental biotechnology for sustainable Resource recovery. Biotechnol reports (Amsterdam, Netherlands) 2020, 28, e00563, doi:10.1016/j.btre.2020.e00563.

>>The aim of the study could be more analytically presented, as well as the section of the conclusions, which could be more comprenhensive. >>

As suggested by the reviewer, we revised Introduction section (lines 76 to 79 of the revised manuscript), and Conclusions section (lines 534 to 546 of the revised manuscript).

> Line 151 check spelling.

Thank you for pointing our mistake. Spelling mistake “wight” was corrected to “weight”.

Thank you again.

Reviewer 3 Report

Title: Stimulation of jasmonate signaling pathway by lipopolysac-2 charide (LPS) from Rhodobacter sphaeroides, a purple 3 non-sulfur bacterium (PNSB), in rice root

Dear Editor and Authors,

This article is part of the research area of the authors, who present the continuation of their research regarding the effect of lipopolysaccharides from PNSB on growth, root development, and gene expression of plants. The authors demonstrate, from their experiments, that indeed the LPS are responsible for the expression of genes. Interestingly, their experiments found genes related to tolerance to biotic and abiotic stress.

Overview and general recommendation

The work is well written and logically organized. In addition, the supplementary material helps to eliminate any possible doubt or questioning of the data. Just consider that some points should be commented or corrected to improve the document.

Minor comments

Please, change all “et al” in the document to “et al.”

Figure 1 is very small, please make it bigger to make it readable. Review the figure captions, it is necessary to eliminate the space, so that it is a single paragraph (the title of the figure and the description) and the text is clearer.

In Fig. 1, it is necessary to correct the text there are errors. For example, it says: “Root development was evaluated by the total root length (Figure 1a and Figure 2a)”, but in fact it is “Root development was evaluated by the total root length (Figure 1a). and Figure 1d)”.

Check the alignment of the cells in the Tables, the information cannot be followed without confusion, since some cells are aligned at the top, others at the center and some below.

In Fig. 4, the Plot of logFC values and expression level is presented. Please, since few upregulated genes are indicated with red circles, put them at the end of the figure description. The same applies for the Figures 5 and 6.

In Figures 4, 5 and 6 it is necessary to add the incise to the graphics, they only appear in the figure captions.

I suggest dividing the Discussion section into subsections. The information that is presented is a lot and it is difficult to read.

The work is well written. 

Author Response

Dear Reviewer,

Thank you very much for your time and effort to review our manuscript, and your quite helpful comments and suggestions to improve our manuscript.

(Comment)

Please, change all “et al” in the document to “et al.”

(Answer)

As suggested by the reviewer, we change all “et al” to “et al.” Thank you for pointing it out.

(Comment)

Figure 1 is very small, please make it bigger to make it readable. Review the figure captions, it is necessary to eliminate the space, so that it is a single paragraph (the title of the figure and the description) and the text is clearer.

(Answer)

As suggested by the reviewer, we made Figure 1 bigger, and revised figure caption.

(Comment)

In Fig. 1, it is necessary to correct the text there are errors. For example, it says: “Root development was evaluated by the total root length (Figure 1a and Figure 2a)”, but in fact it is “Root development was evaluated by the total root length (Figure 1a). and Figure 1d)”.

(Answer)

Thank you for pointing our mistakes. We corrected these errors.

(Comment)

Check the alignment of the cells in the Tables, the information cannot be followed without confusion, since some cells are aligned at the top, others at the center and some below.

(Answer)

Thank you very much for your helpful suggestion. We revised the Tables and corrected the alignment of the cells.

(Comment)

In Fig. 4, the Plot of logFC values and expression level is presented. Please, since few upregulated genes are indicated with red circles, put them at the end of the figure description. The same applies for the Figures 5 and 6.

(Answer)

As suggested by the reviewer, we put red circle at the end of figure descriptions of Figures 4, 5 and 6. Thank you for your suggestion.

(Comment)

In Figures 4, 5 and 6 it is necessary to add the incise to the graphics, they only appear in the figure captions.

(Answer)

We are sorry for forgetting to put the incise in these figures. We added the incises to these figures.

(Comment)

I suggest dividing the Discussion section into subsections. The information that is presented is a lot and it is difficult to read.

(Answer)

Thank you for very helpful suggestion. We divided the Discussion section into subsections. We believe this modification makes the text much easier to read.

Thank you again.

Round 2

Reviewer 1 Report

I agree to accept this manuscript for publication in microorganisms journal because the authors have clearly responded to and revised it.

Author Response

Dear Reviewer,

>> I agree to accept this manuscript for publication in microorganisms journal because the authors have clearly responded to and revised it. >>I

Thank you very much for your quite helpful suggestions.

Best regards,

Hitoshi Miyasaka